# Green Synthesis of Gold, Silver, Copper, and Magnetite Particles Using Poly(tartaric acid) Simultaneously as Coating and Reductant

**DOI:** 10.3390/polym15234472

**Published:** 2023-11-21

**Authors:** Alexander Bunge, Teodora Radu, Gheorghe Borodi, Sanda Boca, Alexandrina Nan

**Affiliations:** 1National Institute R&D for Isotopic and Molecular Technology, 67-103 Donat Street, 400293 Cluj-Napoca, Romaniateodora.radu@itim-cj.ro (T.R.); gheorghe.borodi@itim-cj.ro (G.B.); sanda.boca@ubbcluj.ro (S.B.); 2Interdisciplinary Research Institute in Bio-Nano-Sciences, Babes-Bolyai University, 42 T. Laurian Str., 400271 Cluj-Napoca, Romania

**Keywords:** poly(tartaric acid), gold nanoparticles, silver nanoparticles, copper particles, magnetite nanoparticles, reduction

## Abstract

Poly(tartaric acid) is a relatively recently described polymer that can be easily synthesized and scaled up from a readily available renewable material (tartaric acid). This article demonstrates its use in a green synthesis of gold nanoparticles, silver nanoparticles, copper particles, and magnetite nanoparticles. In this case poly(tartaric acid) acts both as a reductant and as a coating agent. To our knowledge this is the first green synthesis of several different types of nanoparticles using only one reagent (polytartrate) as both reductant and coating. The resulting particles were analyzed by XRD, TEM/SEM, EDX, FTIR, DLS, zeta-potential, XPS, and UV/VIS spectroscopy. Preliminary studies of the thermal behavior of mixtures of different types of particles with poly(tartaric acid) were also conducted. The obtained particles show different sizes depending on the material, and the coating allows for better dispersibility as well as potential further functionalization, making them potentially useful also for other applications, besides the inclusion in polymer composites.

## 1. Introduction

Nanoparticles have been the subject of extensive research in the last two decades. They possess exceptional properties that vary depending on the material they are made of [1]. Due to their high surface-to-volume ratio, they exhibit high activity in chemical reactions and as sorbents and also show specific physical, optic, and magnetic properties that cannot be found in the bulk material of the same composition. Consequently, nanoparticles have found applications in many fields, ranging from cosmetics [2,3] to electronics [4,5] and medicine [6,7].

In the last decade, research on synthesis methods of nanoparticles has taken a new direction due to problems with resource availability, pollution, and the need for a circular economy. Nanoparticles are already widely used in various fields and in large quantities, making efforts to produce them through green synthesis standards essential.

Many research groups have frequently chosen gold nanoparticles over other materials due to their unique properties. There are several reproducible standard methods for synthesizing them [8,9], and they can be easily functionalized through thiol linkage. In addition, gold nanoparticles are biologically inert and have remarkable optical properties, such as surface plasmon resonance [10]. As a result, they have been extensively studied for various applications, including nanomedicine and drug delivery, sensing, catalysis, and energy harvesting [11,12].

Silver nanoparticles are also part of popular research due to their properties, such as easy functionalization [13,14] and surface plasmon resonance. Although they share some similarities with gold nanoparticles, there are also differences that make them attractive in their own right. One major advantage of silver nanoparticles is that they are significantly cheaper than gold precursors, making it possible to use them on a comparatively large scale. Additionally, they are excellent thermal and electric conductors [15,16]. However, it is worth noting that silver nanoparticles are not biologically inert, but possess some antimicrobial activity which makes them interesting for different fields of nanomedicine [17]. Besides that, they have other applications in textile and food industries due to their antibacterial properties, and in electronics due to their conductivity [18].

Copper nanoparticles are not as widely researched as silver and gold nanoparticles, with only 965 articles in the last 5 years containing “copper nanoparticles” in their title compared to 8292 for silver nanoparticles and 6638 for gold nanoparticles [19]. However, copper nanoparticles are still attractive due to their lower cost of precursors, reliable functionalization, excellent thermal and electrical conductivity, and antimicrobial properties. These properties make them useful in fields such as disinfection, catalysis, water depollution, and electronics [20,21,22].

All three types of nanoparticles are usually synthesized, aside from physical grinding of the metals, by reductive processes. The most common chemical reductants are either strong reductants, such as borohydride [23,24,25,26], or milder reductants, such as citric acid [27,28,29]. Most of these methods use an aqueous reaction medium. However, another popular reaction to synthesize metal nanoparticles, particularly silver and copper, is the polyol reaction. In this reaction, a polyol, usually ethylene glycol, serves as the reductant at elevated temperatures, and also as the co-solvent [30]. Moreover, recently, a lot of research has been conducted on preparing these noble metal particles in an ecofriendly way, either by using biological means, such as living or dead cells [31,32,33], or various aqueous extracts from plants, fungi, or bacteria as reductants [34,35].

Magnetite nanoparticles have multiple chemical approaches for their synthesis. While they are not metallic, they are a mixed valence oxide. One can prepare them from iron precursors with appropriate valence, such as co-precipitation [36], hydrothermal synthesis [37,38], microemulsion [39,40], or oxidation from ferrous precursors [41]. Reductive methods can also be used, such as the polyol process [42], thermal decomposition [43,44], or combustion method [45]. To adhere to the green chemistry doctrine, as with the other types of nanoparticles, great efforts have been made to prepare these nanoparticles [46]. Magnetotactic bacteria are the focus of research when it comes to biological synthesis [47]. Alternatively, plant extracts are commonly used to synthesize magnetite nanoparticles [48]. These nanoparticles have unique properties such as superparamagnetism [49,50] and optic properties [51], making them useful in various fields such as electronics/data storage [52], catalysis [53,54], magnetic separation [55], depollution [56], biomedical applications (especially hyperthermia and as MRI contrast agent), and magnetorheological fluids [57].

Poly(tartaric acid) is a polymer that has been recently discovered. It is notable due to the fact that it is made from a green precursor (tartaric acid, which is produced biologically) and is synthesized using a simple thermal treatment. This method of synthesis also allows for easy scalability, making it a promising candidate for future industrial applications [58]. Additionally, the polymer is biocompatible [59] and should easily degrade due to its ester bonds. Despite its potential, poly(tartaric acid) has only been used to coat magnetite nanoparticles in a secondary step after co-precipitation synthesis of uncoated magnetite nanoparticles [58,59]. Nonetheless, it has several advantages as a coating for nanoparticles.

It is green and synthesized in an easy, scalable way, making it useful for applications where larger quantities are necessary, such as sorbents for water depollution, catalysts for industrial scale reactions, or in electronics. It is a polymer with many carboxylic groups, making the coating more stable against exchange, causing a higher stability in suspension, and allowing for easy functionalization via ester or amide bonds. Poly(tartaric acid) has great potential in the fields of catalysis and water depollution, but it also has biomedical applications due to its biocompatibility and potential biodegradability.

As part of our ongoing research on poly(tartaric acid), we aim to investigate its thermal properties when used alone or as a composite where different types of nanoparticles are used as fillers. To achieve this, we require synthesis methods for different types of nanoparticles. While there are several techniques available in the literature, it would be beneficial for our research to have nanoparticles with the same coating that can combine well with bulk poly(tartaric acid), ideally with the polymer itself, to form a homogeneous mixture. Additionally, we need to produce high quantities of nanoparticles to investigate their thermal properties thoroughly.

The novelty of our research consists of the development of a new green method for synthesizing gold, silver, copper, and magnetite nano- and microparticles developed using only water as solvent and poly(tartaric acid) as both the reductant and coating.

The resulting particles possess a uniform coating of the same polymer as the poly(tartaric acid) matrix in which they will ultimately be integrated, rendering them exceptionally advantageous. Furthermore, our procedure yields an ample quantity of silver and magnetite nanoparticles, as well as copper particles, facilitating subsequent testing to determine their efficacy and establish their practicality.

The synthesis of poly(tartaric acid) is achieved using a green method that involves only thermal treatment and does not require solvents. When compared to traditional methods, such as the synthesis of gold nanoparticles using citrate or borohydride, our method offers an immense advantage in that the reducing agent also serves as a polymeric coating agent. Due to the carboxyl groups, the polymer will attach better to the nanoparticle surface due to its multiple carboxylate groups. Moreover, it provides greater stability due to electrostatic and steric repulsion. A polymeric coating is thicker than a single layer of small molecules such as citrate. In contrast to other green methods that use biological systems, either in vivo or as extracts, our methods have a distinct advantage. Only one polymer serves as a coating, rather than a mixture of various small molecules and polymers, as would be the case with an undefined medium such as a plant extract.

The particles were subjected to a characterization process, followed by preliminary tests to assess the thermal conductivity, diffusivity, and specific heat of silver, copper, and magnetite particles. The results of these tests are of importance to ascertain the suitability of these particles for various applications, especially in the electronics industry. The findings of this study may provide valuable insights into the properties of these materials, which can further contribute to the development of advanced materials for use in various industrial fields.

## 2. Materials and Methods

### 2.1. Materials

Metallic gold (99.99%) was a donation from Dr. Lucian Barbu-Tudoran, Cluj-Napoca. Potassium permanganate, sodium hydrogen carbonate (anh.), and ferric chloride hexahydrate were obtained from Alfa Aesar. Hydrochloric acid (37%) and DL-tartaric acid were obtained from Merck. Sodium carbonate (anh.) was received from SC Simex Srl. (Bucharest, Romania). Copper sulfate pentahydrate was obtained from Reactivul Bucuresti (Bucharest, Romania). All chemicals were used without further purification.

### 2.2. Syntheses

#### 2.2.1. Tetrachloroauric Acid Solution

The aqueous solution of tetrachloroauric acid was prepared according to a previously published procedure [60]: potassium permanganate (6 g) was placed in a 100 mL two-neck flask with a dropping funnel and connected using a gas inlet with a 250 mL three-necked flask heated using an oil bath. The dropping funnel was charged with concentrated hydrochloric acid (30 mL), and metallic gold (586 mg) and water (200 mL) were placed in the three-necked flask. A tube connected this flask to an empty gas trap and then further into an Erlenmeyer flask filled with a solution of sodium thiosulfate (2.56 g) in water (100 mL). The oil bath was heated to ca. 55 °C and hydrochloric acid was added dropwise. After 4 h of stirring, all gold was dissolved and the tetrachloroauric acid stored in the fridge until use.

#### 2.2.2. Poly(tartaric acid) **1**

Poly(tartaric acid) was synthesized according to a previously published procedure [58]: tartaric acid (11.428 g) was ground into a fine powder using a mortar and put into a 400 mL wide beaker. The beaker was then put into an oven and heated at 160 °C for 6 h. Near the end of the reaction, the mixture rose in the beaker, forming bubbles. After cooling down to room temperature, 9.566 g of a tan solid were obtained.

The poly(tartaric acid) **1** (1 g) was titrated in distilled water at ca. 50 °C and it was found that approximately 10 mmol of sodium hydroxide are necessary for neutralization to pH = 7.

#### 2.2.3. Synthesis of Gold Nanoparticles **2**

Poly(tartaric acid) **1** (100 mg) was dissolved with warming in distilled water (100 mL) by addition of 10 M NaOH (100 µL), then heated to 80 °C and stirred. Tetrachloroauric acid solution (see above, 7 mL) was then added at once and the solution stirred at that temperature for another 12 min. The deeply purple-colored suspension was then centrifuged and washed with distilled water (2×) before being stored as aqueous suspension.

#### 2.2.4. Synthesis of Silver Nanoparticles **3**

Poly(tartaric acid) **1** (3.0 g) was dissolved in distilled water (50 mL) with the help of sodium hydroxide (1.214 g), and 100 mL of a 0.1 M AgNO_3_ solution was added at once. The mixture was stirred under reflux for 2 h and afterwards centrifuged and washed with distilled water (2×) and acetone. After drying over night at 60 °C, 547 mg of silver nanoparticles **3** were obtained.

#### 2.2.5. Synthesis of Cuprous Oxide Particles **4a**

Copper sulfate pentahydrate (5.00 g) was dissolved in distilled water (20 mL) and added to a solution prepared from poly(tartaric acid) **1** (6.0 g), sodium hydrogen carbonate (6.0 g), and distilled water (200 mL) under heating. The mixture was then stirred at 80 °C for 5 h, during which time sodium carbonate (1 g) was added portionwise. The resulting particles were centrifuged, washed with distilled water, and dried to obtain 49 mg **4a** of red powder.

#### 2.2.6. Synthesis of Cuprous Oxide/Copper Particles **4b**

Sodium hydroxide (800 mg) was added to a stirred solution of copper sulfate pentahydrate (3.50 g). The precipitate was centrifuged and aged for 4 days in distilled water, after which the supernatant was decanted and the wet precipitate added with distilled water (10 mL) to a solution of poly(tartaric acid) **1** (3.0 g) and sodium hydroxide (1.2 g) in water (40 mL). The mixture was then heated in a 100 mL Teflon-lined autoclave at 200 °C for 6 h, with the resulting particles centrifuged and washed with distilled water (2×) before being dried over night at 60 °C. A wine-red powder (633 mg) of **4b** was obtained.

#### 2.2.7. Synthesis of Cuprous Oxide/Copper Particles **4c**

A solution of 10 M NaOH (4 mL) was added to a stirred solution of copper sulfate pentahydrate (5.00 g), and then a solution of poly(tartaric acid) **1** (3.0 g) and sodium hydroxide (800 mg) in distilled water (20 mL) was also added. The deep blue solution was transferred into a 100 mL Teflon-lined autoclave and heated at 200 °C for 16 h. The resulting particles were centrifuged, washed with distilled water (2×) and acetone, and dried at 60 °C overnight to obtain **4c** (1.04 g) of a wine-red powder.

#### 2.2.8. Synthesis of Copper Particles **4d**

A solution of 10 M NaOH (5 mL) was added to a stirred solution/suspension of copper sulfate pentahydrate (2.50 g) and poly(tartaric acid) **1** (3.0 g) in water (40 mL) under warming at ca. 50 °C. The deep blue solution was transferred into a 100 mL Teflon-lined autoclave and heated at 200 °C for 72 h. The resulting particles were centrifuged, washed with distilled water (2×) and acetone (2×), and dried at 60 °C overnight to obtain **4d** (563 mg) of a red powder.

#### 2.2.9. Synthesis of Ferrous Tartrate Particles **5a**

Sodium hydroxide (1.2 g) was added to a solution/suspension of ferric chloride hexahydrate (2.70 g) and poly(tartaric acid) **1** (3.0 g), and the ochre-colored mixture stirred for 1 h before being transferred to a 100 mL Teflon-lined autoclave. The autoclave was heated at 200 °C for 14 h and the white precipitate centrifuged, washed with distilled water (3×), and dried overnight at 60 °C to obtain 1.47 g **4a** of an off-white powder.

#### 2.2.10. Synthesis of Magnetite Nanoparticles **5b**

Sodium hydroxide (1.6 g) was added to a solution/suspension of ferric chloride hexahydrate (2.70 g) and poly(tartaric acid) **1** (1.0 g), and the ochre-colored mixture stirred for 1 h before being transferred to a 100 mL Teflon-lined autoclave. The autoclave was heated at 200 °C for 72 h and the resulting black solid was separated magnetically before being washed with distilled water (3×) and dried overnight at 60 °C to obtain **4b** (773 mg) of a black solid.

#### 2.2.11. Preparation of Mixtures of Particles with Poly(tartaric acid) **6**

A mixture of poly(tartaric acid) **1** (1.80 g) and particles (200 mg) was mixed and finely ground in a mortar. Thus, sample **6a** was obtained using nanoparticles **3**, **6b** with **4d**, and **6c** using **5b**. The samples were then pressed into pellets for thermal analysis.

### 2.3. Characterization

The size and shape of the nanostructures were examined by scanning transmission electron microscopy (STEM) with a Hitachi HD2700 equipped with a cold field emission gun, Dual EDX System (X-Max N100TLE Silicon Drift Detector (SDD)) from Oxford Instruments (Abingdon, UK). For the analysis, a suspension of the samples was sonicated (<10 s) with a UP100H ultrasound finger and deposited by the droplet method on a 400-mesh copper or nickel grid coated with a thin carbon layer. For both types of analysis, the nominal operating tension was 200 kV. The size of the nanoparticles was determined using the ImageJ software.

Fourier transform infrared (FTIR) spectra were recorded using a JASCO FTIR 4600A spectrophotometer with ATR-PRO-ONE accessory, CO_2_−, H_2_O−, ATR−, and baseline-corrected, as well as smoothed and normalized for better visibility of the bands.

UV–VIS spectra were taken in aqueous suspension using a Jasco V-550 UV–VIS Spectrophotometer (JASCO Deutschland GmbH, Pfungstadt, Germany) equipped with a double-beam photometer and a single monochromator, using 10 mm length quartz cells. After the particles were suspended in water, a drop of diluted ammonia was added for stabilization and the sample dispersed using a UP100H ultrasound finger.

Most X-ray power diffraction (XRPD) measurements were performed with a Bruker D8 Advance X-ray diffractometer (Billerica, MA, USA), with a Ge (111) monochromator for Cu-Kα1 radiation (λ = 1.5406 Å) having the source power of 40 kV and 40 mA, at room temperature and LynxEye position-sensitive detector. Samples **4c** and **4d** were measured on a Smart Lab Rigaku diffractometer with Cu-K*α* radiation at room temperature.

Zeta potential, as well as the size distribution via dynamic light scattering (DLS), were measured at 25 °C using the Zeta sizer NanoZS90 from Malvern Panalytical Ltd., Malvern, UK. Analysis was performed at a scattering angle of 90° and a temperature of 22 °C. All experiments were performed in triplicates and the data are expressed as mean ± standard deviation (SD). The samples were ultrasonicated for 2 min and homogenized before each measurement using a test tube shaker (IKA) at a fixed speed of 2800 rpm.

A SPECS XPS spectrometer equipped with an Al/Mg dual-anode X-ray source, a PHOIBOS 150 2D CCD hemispherical energy analyzer, and a multichanneltron detector with vacuum maintained at 1 × 10^−9^ Torr was used to record the XPS spectra. The Al K_α_ X-ray source (1486.6 eV) was operated at 200 W. The XPS survey spectra were recorded at 30 eV pass energy and 0.5 eV/step. The high-resolution spectra for the individual elements were recorded by accumulating 10 scans at 30 eV pass energy and 0.1 eV/step. Data analysis and curve fitting was performed using CasaXPS software with a Gaussian–Lorentzian product function and a nonlinear Shirley background subtraction. Peak shifts due to any apparent charging were normalized with the C1s peak set to 284.8 eV. The high-resolution spectra were partly deconvoluted into the components in order to determine the particular bond types present at the sample surface.

The thermal conductivity was measured in a Hot Disk TPS 2500S (Hot Disk AB, Kagaku, Gothenburg, Sweden) apparatus using a 5464F1 sensor using the transient plane source (TPS) method. The equipment includes determining the diffusivity and volumetric heat of materials. The method principle consists of applying a short heat pulse of a predetermined duration to the sample, initially maintained at thermal equilibrium using a TPS sensor with a double function: constant heat source and temperature sensor placed between two identical samples. The transient temperature response of the samples was recorded and further used to estimate the thermal conductivity. To obtain results with excellent accuracy, the samples were prepared in the form of identical pellets with a radius of 1 cm and a thickness of about 4 mm to ensure that a TPS sensor with a diameter of 2 mm could be used. In this way, the heat generated by the spiral area does not diffuse to the sample outside boundary within a predefined period of measurement time.

## 3. Results and Discussion

Poly(tartaric acid) **1** was synthesized according to known procedures [59]. The polymerization process of tartaric acid occurs through an auto-catalyzed esterification that results in the elimination of excess water. This reaction involves a tiny amount of decarboxylation, a process that has been already described in the literature [58]. The resulting polymer can show motifs of linear polyesters, but since more than two hydroxy and carboxyl groups exist, a lactide formation as well as branched polymerization is also possible (Figure 1). Under the reaction conditions used, decarboxylation is not significant, so decarboxylation products were not included in the formula of **1**.

Poly(tartaric acid) **1** is not completely soluble in water; however, if the solution is neutralized and gently heated, it does dissolve. During the reactions to form particles as described below, part of **1** will get oxidized and thus degraded and/or split in smaller units or even monomers. The reason why it can still serve as a coating agent is that, if the degradation is not complete, a competition between tartrate or the degradation products and polytartrate to attach to the particle surface will occur. Due to the chelate effect—an entropic effect that predicts that a complexation will occur more likely with one multidentate ligand than with multiple monodentate ligands, because the final ensemble of molecules will have higher degrees of freedom—even in this mixture the complexation will occur with polytartrate, rather than with the degradation products.

### 3.1. Synthesis of Gold Nanoparticles ***2***

Poly(tartaric acid) **1** was used as both a reducing and coating agent in the synthesis of gold nanoparticles. Gold, being a noble metal, can be reduced even by weak reductants under mild conditions. This makes gold nanoparticles the easiest to form among the desired types of particles. We employed conditions similar to those used in the Turkevich synthesis method to synthesize polytartrate-coated gold nanoparticles. The method involved adding tetrachloroauric acid, prepared from metallic gold by oxidation with chlorine, to a heated solution of sodium polytartrate (Appendix A). After two minutes, a discoloration of the solution demonstrated that the reaction had taken place and, after 12 min, the mixture had a deep violet color typical of gold nanoparticles. Unlike the other particles described here, gold nanoparticles **2** were kept in aqueous suspension.

### 3.2. Synthesis of Silver Nanoparticles ***3***

Silver has a lower reduction potential than gold. Therefore, following the same reaction conditions as for the synthesis of gold nanoparticles may lead only to a partial reduction to metallic silver. However, if the reaction time is prolonged to two hours, silver nanoparticles can be obtained. The reaction mixture changes its color from brown to gray, indicating the formation of silver nanoparticles. Hence, by slightly modifying the reaction conditions, it is possible to synthesize silver nanoparticles **3**.

### 3.3. Synthesis of Copper Containing Particles ***4***

The reaction for copper was different from that of silver due to the lower reduction potential of copper. Despite boiling the solution for a long time, the color remained blue due to a copper–polytartrate complex. After adding more base, a small amount of red solid formed (**4a**), indicating that the reduction did not reach completion and only Cu_2_O was produced. This is different from other reports where copper nanoparticles were formed at lower temperatures using glucose [61], ascorbic acid [62], or plant extracts [63] as reductants. While the mechanism of the reduction using polytartrate is not entirely clear, it likely involves partial hydrolysis of the polytartrate to tartrate ion and subsequent oxidation, possibly under decarboxylation. Mild reductants such as glucose, ascorbate, and plant extract flavonoids are more effective than tartaric acid. For instance, glucose has an aldehyde group that can be oxidized, while the oxidation product of ascorbate is more stable; in plant extracts, flavonoids are the primary agents responsible for the reduction, as they are excellent reductants. Since polytartrate is a polyol, it was believed to work similarly to a polyol process at higher temperatures (except using the polyol only as a reagent, not as solvent). Thus, three more experiments were conducted under hydrothermal conditions for 6 h, 16 h, and 72 h to obtain particles **4b**, **4c**, and **4d**. The supernatant was no longer blue, indicating that no cupric ions remained in the solution, and the mass of the products obtained was significantly higher than in the case of **4a**. To determine whether the particles obtained were fully reduced copper or only partially reduced cuprous oxide, XRD was used (see Section 3.5).

### 3.4. Synthesis of Iron-Containing Particles ***5***

In comparison to copper, ferric salts have a lower reduction potential. This makes it very challenging to produce zerovalent iron particles through a polyol process. Instead, magnetite is obtained almost exclusively [30]. Due to the success of copper nanoparticles synthesis, the reaction to obtain particles **5** was carried out under hydrothermal conditions. Lower temperature reactions were not endeavored. In the first attempt (**5a**), no magnetic material was produced, only a considerable amount of off-white powder. Since the supernatant was not highly colored, it was likely that product **5a** contained iron. Since it was almost colorless, the iron was likely in a completely reduced form (ferrous salt) rather than being mixed valence like magnetite. To synthesize **5b**, the quantity of polytartaric acid was reduced. However, the amount of tartrate would still be more than the stoichiometric amount needed for the reduction of ferric ions to magnetite upon full hydrolysis. Otherwise, no polytartrate could remain to form a coating. The reaction time was also prolonged. Under these conditions, it was possible to obtain the black, magnetic product **5b**.

### 3.5. Phase Analysis by XRD

Samples **2**–**5** were analyzed by XRPD (Figure 1 and Appendix A). For sample **3**, it can clearly be seen that it consists of metallic silver, hence the reduction was successful. In the case of **4**, under reflux conditions (**4a**) only cuprous oxide was found, while under hydrothermal conditions (**4b**–**d**) successively more metallic copper could be identified in the diffractograms. In order to achieve complete reduction to copper, particles using only poly(tartaric acid) **1** as both coating and reductant, hydrothermal conditions (200 °C), and a reaction time of 72 h are thus needed. For the reduction of ferric chloride, Appendix A shows that **5a** consists mainly of ferrous tartrate, which agrees with the observations made during synthesis (light color, not magnetic). In the diffractogram, there are some reflections stemming from an impurity, tartaric acid (PDF 40-0609), such reflections having been observed previously for the synthesis of ferrous tartrate [64]. For **5b**, however, the diffractogram points clearly to a spinel structure, which makes it likely that magnetite was formed (maghemite would show the same diffraction pattern, but the black color of **5b** makes magnetite more likely).

### 3.6. Morphology and Elemental Composition by TEM/SEM and EDX

The morphology of the obtained samples was determined by TEM and SEM (Figure 2, Appendix A). For sample **2**, one can see that the gold nanoparticles have been formed, with a mean size of 35 ± 11 nm. The sample thus appears to be relatively polydisperse, and also the shapes are not uniform, from quasi-spherical to triangular, pentagonal, or hexagonal. That the nanoparticles consist of metallic gold can be inferred from the dark color of the particle, pointing to a high density and, thus, to a particle consisting only of metal and not lighter elements such as oxygen. In Figure 2a, it is also visible that the nanoparticles are coated with a polymer (ca. 4 nm thickness). Sample **3** consists of nanoparticles between 25–400 nm, where most particles are in the sub-100 nm range and only a few are larger. The shape of particles **3**, similar to **2**, consists of different types, from quasi-spherical to hexagonal. From Appendix A, again, a polymer coating of ca. 3 nm can be seen.

The cuprite and copper particles **4a**–**d** are also relatively polydisperse, with a particle size of 270 ± 168 nm in the case of **4a**, 2.58 ± 1.05 µm in the case of **4b**, 3.39 ± 2.06 µm for **4c**, and 723 ± 372 nm for **4d**. The shapes of **4a** are irregular; however, for **4b**, most particles appear to be either fully formed or truncated/broken octahedra. In the case of **4c** and **4d**, a mixture of unshaped and octahedral shapes was noticeable. It is interesting that the particle size for **4d** significantly shrinks, which could be a sign that during the reaction a dissolution–reprecipitation mechanism occurs to form the final copper particles.

An attempt was made to use EDX to show the differences in composition of samples **4a**–**d** (Appendix A). In all samples of particles, copper was found to be the predominant element, although the oxygen content was also in all cases insignificant, unlike it was expected at least in the case of **4a**. In the case of **4b** and **4c**, this could possibly be explained by the fact that just particles consisting of metallic copper were measured, a fact that is supported by the observation that the dead time while measuring EDX on the particles was higher than for other areas of the carbon grid (among other reasons, this happens when a sample is conductive). Since the TEM grid consists of carbon, most of the detected carbon in EDX generally stems from the grid, and thus carbon content cannot usually be used to evaluate the sample. In the cases of samples **4**, however, the carbon content was found to be unusually low. The two reasons for this are, first, that, because of the large size of the copper particles, the carbon part of the grid was not detected by EDX and, second, that the organic coating was quite low compared to the inorganic core (even with a coating of several nm, it would be little compared to a µm sized particle).

The ferrous tartrate particles **5a** appear to be mainly broken rods of 793 ± 555 nm length. This is also consistent with previously obtained results under similar conditions [64]. In contrast, sample **5b** consists of spherical nanoparticles of 18 ± 4 nm. A coating was not visible in TEM; however, a comparison between the same nanoparticles in SEM and TEM showed that particles in SEM appear ca. 10% larger, which can be at least an indication that a coating exists.

### 3.7. Compositional Analysis Using FTIR

In order to better analyze the composition of the prepared materials, FTIR measurements were conducted for samples **3**, **4**, and **5** (Figure 3 and Appendix A). Additionally, spectra for poly(tartaric acid) **1** and its sodium salt **7** were recorded. The spectrum of poly(tartaric acid) **1** corresponds well to previously recorded spectra [58]. Metallic samples such as silver and copper generally show no bands in the IR spectrum within the range measured. This is why FTIR can only be indirectly used to determine whether particles **3** and **4** consist of metallic silver/copper. Silver and cuprous oxide show a specific band around 600 cm^−1^ in FTIR [65,66]. For samples **3** and **4d**, this band is not visible, proving that both samples consisted of metallic silver/copper. In contrast, samples **4a**–**c** each show this characteristic band, indicating that they consisted at least partially of cuprous oxide. The band decreases in relative intensity from **4a** to **4c**. Both of these findings correlate well with the results from XRD. Sample **5a** corresponds well to literature data of ferrous tartrate (broad band at 3200–3500 cm^−1^, bands at 1589, 1412, 1230, 1084, 1043, 742, and 633 cm^−1^) [67], a fact which confirms the analysis conducted by XRD. Sample **5b** shows a band at 532 cm^−1^, which can be assigned to the ν(Fe-O) of magnetite. XRD alone cannot distinguish between the two spinel iron oxides magnetite and maghemite, but the absence of a band > 635 cm^−1^ proves that the particles **5b** are indeed magnetite and not maghemite [36,68,69]. Aside from composition of the core particles, FTIR spectroscopy can often be used to determine whether or not the inorganic particles are coated. In the case of **5a**, the strong bands of the core (ferrous tartrate) prevent this assessment; however, in all other cases, it can be seen that the particles are coated in polytartrate. A sample of sodium polytartrate **7** was prepared by neutralizing poly(tartaric acid) with sodium hydrogencarbonate, and its IR spectrum taken for comparison with the spectra of the particles. Sodium polytartrate is better suited for this comparison than the free poly(tartaric acid) **1**, as the coating on the surface of particles will most likely be in anionic form instead of occurring as the free polyacid. Broad bands between 3000–3500 cm^−1^ can be seen in samples **3** and **5b**, corresponding to ν(O-H) of polytartrate and of inorganic hydroxy groups on the surface of the inorganic core. In samples **4**, this band is less visible because of the large size of the particles, corresponding to a smaller surface-to-volume ratio and thus a smaller hydroxylated surface. Bands at ca. 2850 cm^−1^, 2920 cm^−1^, and 2950 cm^−1^ correspond to ν(C-H) of the poly(tartrate). Sodium polytartrate **7** shows a band at 1735 cm^−1^ which is the ν(C=O) of free acid groups. A band at a similar wavenumber (~1695–1712 cm^−1^) can also be found in samples **3**, **4c**, **4d**, and **5b**, but not in **4a** and **4b**, a phenomenon which could indicate that all carboxylate groups are bound to the surface in these two samples, whereas, in the other samples, there exist also some free carboxylic acid groups. The free carboxylic acid groups can also be a sign that the surface coating consists of polytartrate and not monomeric tartrate, because, due to the chelate effect (an entropic effect), all carboxylic acid groups should participate in the coating, if possible. If this does not happen, the reason would be electrostatic (too many negative charges in one molecule are energetically unfavorable) or steric (one side of the molecule is bound to the surface of the particles, while the other is blocked by other molecules and points away from the surface). Both of these reasons would be more probable for a polymer than for a monomer, since the former is bigger. The carboxylate and ester ν(C=O) of **7** are at 1599 cm^−1^ (symmetric) and 1386 cm^−1^ (asymmetric) and can be found in the same region for **4a**, **4c**, **4d**, and **5b**, while for **4b** the bands are not visible (likely because of the large size of the particles) and for **3** they are shifted (1499 cm^−1^ and 1343 cm^−1^). The reason for this shift is unknown. Thus, the fact that all particles show a coating can be confirmed also by FTIR, even though the relative amount of it is smaller in the large particles (e.g., **4b**). The coating is most likely polytartrate.

### 3.8. UV/VIS Analysis of Samples ***2***, ***3***, and ***5b***

UV/VIS spectroscopy can offer important information about different characteristics of nanoparticles, including shape and size. This is an especially important method of characterization for gold nanoparticles, as they are generally produced in small amounts and, thus, most other characterization methods are not feasible for them. Gold, and to an extent also silver nanoparticles, demonstrate surface plasmon resonance, which results in a strong absorbance in the visible region (ca. 500–600 nm for gold nanoparticles) [10,70]. Only nanoparticles **2**, **3**, and **5b** were measured this way, as they were sufficiently stable in suspension (Figure 4).

In the visible light range, gold nanoparticles **2** show a band at 529 nm together with a broad band at ca. 650 nm. By comparison with spectra of gold nanoparticles of different sizes [71], it is possible to determine that that the size of the particles is ca. 40 nm, a size that agrees well with the diameter determined by TEM. The broad band at 650 nm, together with the results from TEM, indicate that some parts of the nanoparticles are also aggregated. Another possibility could be a bidisperse distribution with a significant amount of large-sized (~150 nm) particles, but there was no evidence for this in TEM.

The average size of silver nanoparticles **3** could not be well determined by TEM due to their polydispersity, or by XRD due to the large maximum size of the particles. By comparison with UV/VIS spectra of other silver nanoparticles [72], the average size of the majority of particles in **3** can be determined to be between 40 nm and 50 nm. A band for the population of larger nanoparticles is not visible because of their lower abundance compared to the smaller particles.

Magnetite nanoparticles **5b** show a spectrum with very few distinct bands, with only two broad shoulders of low intensity at ca. 250 nm and 375 nm being distinguishable. A spectrum like this is typical for magnetite [73]. It can be seen, however, that magnetite, and not maghemite, was formed, as the UV/VIS spectrum of maghemite shows distinct differences (a comparatively much stronger increase in absorbance below 400 nm) [74].

### 3.9. DLS and Zeta Potential Measurements

Aside from electron microscopy, dynamic light scattering (DLS) can also be used to determine the size of the synthesized particles. The hydrodynamic diameter and polydispersity index (PDI) of samples **3**, **4d**, and **5b** are shown in Table 1.

Compared to the sizes determined from electron microscopy (Appendix A), the diameter determined by DLS is larger. To an extent, this is normal, as the hydrodynamic diameter determined by DLS also takes into account the shell of water molecules that forms around each particle. This shell increases in size with increasing nanoparticle charges and, with a polyanionic coating such as polytartrate (or, for example, polyacrylate [75,76]), which attracts a larger shell of water molecules, it can be several times larger than the actual particle size. In this case, however, the differences between the measured hydrodynamic diameters and the diameters determined from electron microscopy are big enough that it is likely that at least some aggregation of nanoparticles happened as well. PDIs higher than 0.1 indicate that all three samples are also polydisperse. In the case of **3** and **4d**, these results confirm the observations made from electron microscopy measurements. It is interesting that also the sample **5b** is shown as polydisperse here, which is most likely due to some aggregation, as stated above.

The zeta potential was also measured and is shown to be negative for all three samples (Table 1). Similar values between −30 mV and −40 mV have been obtained for other particles coated with polycarboxylates (polyacrylate) [77,78,79], confirming earlier observations that each type of particles has a definite organic coating of polytartrate.

### 3.10. Surface Analysis by XPS

XPS analysis was conducted for samples **3**, **4d**, and **5b** in order to obtain additional insights into both the inorganic core as well as the organic coating of the samples. The Ag3d, Cu2p, Fe2p, C1s, and O1s core level spectra are shown in Figure 5 and Appendix A.

The Ag3d spectrum for **3** is split in 3/2 and 5/2 peaks, each of which only consists of one component (binding energies 373.9 eV and 367.9 eV), for metallic silver [80]. This confirms the findings from XRD, that metallic silver nanoparticles have been successfully synthesized. For **4d**, the Cu2p_3/2_ peak is made up of several contributions, at 931.4 eV and 933.6 eV, respectively, the former being that of metallic copper and the latter that of cupric ions [81]. The presence of the shake-up satellite structures observed in the Cu2p XPS spectrum was an additional indication of the presence of CuO species in the surface layer. XPS is a surface analysis method, meaning that its penetration depth is relatively low. Since the diffractograms of XRD, which is a bulk investigation technique, for **4d** failed to show any other phase but metallic copper, this means that the cupric ions exist only in small quantities on the surface. The Fe2p_3/2_ spectrum for **5b** can be split into the typical contributions of tetrahedral, octahedral ferric ions and ferrous ions for magnetite. The ratio of Fe^2+^/Fe^3+^ with 0.42 is lower than normal for magnetite (0.5), a phenomenon which can be attributed to some surface oxidation or incomplete reduction due to a lower amount of reductant (polytartrate) in the reaction conditions. If there was a significant amount of maghemite present, however, then this would have been visible in the FTIR spectra. Thus, it can be concluded that only a small amount of the magnetite core is oxidized on the surface. The C1s core spectra for all three samples show contributions at ca. 285.8 eV and 288.2 eV, in addition to a peak at 284.7 eV stemming from C–C impurities on the sample from possible contamination during sample handling. The first two peaks can be assigned to C–O and C=O (from carboxylate), respectively, stemming from the surface coating of polytartrate. For sample **5b**, a small additional peak observed at 289.9 eV is most likely from carbonate. Carbon dioxide forms during the reduction of ferric to ferrous ions under partial decomposition of tartrate and can react with iron ions to form iron carbonate. This iron carbonate (which is not present in significant amounts, or it would be detectable in the XRD diffractograms) is, thus, an indicator of the previously formulated potential mechanism of reduction of metal ions by cleavage of tartrate under formation of carbon dioxide.

The O1s core level spectra (Appendix A) mainly confirm the conclusions drawn from evaluating the C1s spectra. Peaks at ca. 532.6 eV and 531.6 eV stem from C=O (carboxyl) and C–O oxygen, respectively, which again prove the presence of a coating of polytartrate on the particles. For sample **5b**, the peak at 529.6 eV can be attributed to Fe–O bonds, while, interestingly, there is no equivalent peak in the O1s spectrum of **4d**. This means that, despite the presence of Cu^2+^ as shown from the Cu2p spectra, the corresponding Cu–O bonds are not visible, pointing towards a comparatively large amount of organic coating and/or to a relatively small amount of Cu^2+^.

### 3.11. Thermal Conductivity, Diffusivity, and Volumetric Heat

The particles **3**, **4d**, and **5b** were then used in mixtures with poly(tartaric acid) **1** for preliminary tests of thermal behavior of these mixtures. The samples are **6a** for a mixture of poly(tartaric acid) **1** with 10% **3**, **6b** for a mixture with 10% **4d**, and **6c** for a mixture with 10% **5b** (Table 2). The data for thermal conductivity λ at room temperature as well as 50 °C are shown in Figure 6a and Appendix A. It can be seen that λ does not change substantially between the pure polymer **1** and upon mixture with 10% of any of the particles. Upon heating to 50 °C, there is no consistent increase or decrease; however, generally, the changes are very low. In general, inorganic, and especially metal, particles possess a high thermal conductivity compared to organic polymers. Addition of these particles to the polymers as a filler improves thermal conductivity in many cases [82]; however, there are also reports of the addition of a filler decreasing thermal conductivity [83]. This effect is normally attributed to a decrease in the order/crystallinity in the polymeric matrix. Such an effect seems to be occurring here as well, where any increase in thermal conductivity by the filler is (over)compensated by a loss of order. An additional indicator for this could also be the behavior of the polymer **1** and composites **6** at higher temperature. It had originally been planned for the thermal conductivity to be measured also at 100 °C; however, while for poly(tartaric acid) **1** this measurement could be performed, for the composites **6** the measured values were deemed inaccurate because the pellets melted to varying degrees at that temperature. A decrease in the melting point is an indicator of a decrease in crystallinity of a polymer blend [84] and, thus, would lead to a lower thermal conductivity. The thermal diffusivity α is connected to the volumetric heat capacity s and thermal conductivity λ by the formula α = λ/s. With similar thermal conductivity for the measured composites and temperatures, α is roughly inversely proportional to s. The specific heat capacity c (and, thus, also the volumetric heat capacity s, if we assume the density between the samples does not vary by a large amount) can be temperature-dependent to a much higher degree than the thermal conductivity [85] and, indeed, it has been found to be the case here as well (Figure 6, Appendix A). The volumetric heat capacity s decreased with increasing temperature in all cases. Specifically for sample **6a**, s was very large relative to the pure polymer **1**. The other composite samples **6b** and **6c** showed less of a difference.

From these preliminary tests, it has been found that, for poly(tartaric acid) **1**, creating a composite with the particles synthesized here likely lowers the crystallinity and, as a result, the thermal conductivity is lowered instead of raised compared to the pure polymer **1**. Thermal diffusivity and volumetric heat capacity change to a much higher degree between composites and depending on the temperature relative to thermal conductivity. Further investigations using different particle sizes and particles with the same cores but different coatings and different ratios for the composites are needed to verify the mechanism of variation of thermal conductivity of the obtained composites.

## 4. Conclusions

Poly(tartaric acid) was found to be a versatile compound that has the potential to serve as both a coating agent and a reductant in environmentally sustainable processes for the synthesis of gold, silver, copper, and magnetite nanoparticles. The use of poly(tartaric acid) as a coating agent and reductant presents a promising green alternative that is both cost-effective and efficient. The reaction conditions depend on the type of particles produced, ranging from reflux to hydrothermal conditions.

The obtained samples have been investigated by XRD, TEM/SEM, EDX, FTIR, UV/VIS, XPS, DLS, and zeta potential measurements. Following usage of electron microscopy and UV/VIS, it has been found that the particles vary in size from 10 nm to several µm depending on the starting cations added in the reaction mixture. DLS shows that some agglomeration may occur for the particles, mainly for **5b**. This might be because the sample **5b** is magnetic. Efficient coating of the samples was also achieved, with a coating thickness of several nm (TEM). Besides TEM, the presence and identity of the coating (polytartrate) was also proved by FTIR and XPS. The carboxyl groups of the coating provide benefits in terms of particle stabilization (through electrostatic repulsion) in solution (as shown by a zeta potential with values between −30 mV and −40 mV) and surface functionalization (e.g., through EDC coupling). Changing the reaction conditions can lead to a partial reduction (in the case of cuprous oxide particles **4a**) or even to an overreduction (in the case of ferrous tartrate particles **5a**).

Initial tests were conducted to examine the thermal behavior of mixtures containing poly(tartaric acid) **1** and nanoparticles or particles. The results showed that the composites formed possess a thermal conductivity ranging between 0.2 W/mK and 0.3 W/mK, both at room temperature and 50 °C. The slight decrease in the composite thermal conductivity may be attributed to reduced crystallinity compared to that of pure polymer **1**.

This publication represents a pioneering effort in systematically describing chemical synthesis methods that employ a green reagent as both a reductant and a coating agent to produce diverse types of particles, including Au, Ag, Cu, and Fe_3_O_4_. Of relevance to our group is the application of these particles in polymer composites, although the potential to generate highly functionalized and functionalizable particles using green, scalable syntheses is anticipated to have far-reaching implications across several domains. In particular, the fields of catalysis and water depollution stand to benefit significantly from this innovative approach. To the best of our knowledge, this is the first publication that has documented these methodologies in detail.

## Data Availability

The data presented in this study are available on request from the corresponding author.

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
