# Peer review of "Green Synthesis of Gold, Silver, Copper, and Magnetite Particles Using Poly(tartaric acid) Simultaneously as Coating and Reductant"

_polymers, 2023, doi:10.3390/polym15234472_

Round 1

Reviewer 1 Report (Previous Reviewer 1)

Comments and Suggestions for Authors

The manuscript entitled "Green synthesis of gold, silver, copper and magnetite particles using poly(tartaric acid) simultaneously as coating and reductant" aims to demonstrate the use of poly(tartaric acid) in green synthesis of gold, silver, copper, and magnetite nanoparticles. Poly (tartaric acid) was used as both a reductant and coating agent. The obtained particles were analyzed by XRD, TEM/SEM, EDX, FTIR, and UV/VIS spectroscopy. Preliminary studies of the thermal behavior of mixtures of different types of particles with poly(tartaric acid) were also conducted. The topic is interesting and fits well with the scope of the journal. The results are significant and valuable. The authors made an effort to improve the manuscript, but there are still some issues that need to be addressed.

While there are some additions, the introduction is still confusing and not focused enough. It requires substantial alterations. The novelty and motivation for the manuscript are also not apparent and should be more precise.

The results presentation and discussion are improved, but they are still confusing.

Line 288: "as well as a certain amount of decarboxylation" – what does a certain amount of decarboxylation mean? It is not precise enough. Please rephrase.

Sections 3.1. to 3.4. are confused and should be carefully rewritten. The authors added some text, but these paragraphs should be rewritten entirely to be more straightforward. 

Comments on the Quality of English Language

Moderate changes are required. 

Author Response

Dear Reviewer,

     On behalf of the manuscript's co-authors with the code number ID: polymers-2727253, I would like to thank you for the detailed evaluation. We have carefully studied all the helpful remarks and made corrections, which we hope will meet your approval. We appreciate your efforts to provide valuable comments to improve the quality of our manuscript. Please find enclosed the revised manuscript wherein all comments are considered. As a result, of the reviewers’ suggestions, we have rewritten the introduction, large parts of the discussion section as well as the conclusion part. The changes are indicated using the Track Changes option in the revised manuscript. We believe that the paper has improved considerably and become more comprehensive and extended.

The manuscript entitled "Green synthesis of gold, silver, copper and magnetite particles using poly(tartaric acid) simultaneously as coating and reductant" aims to demonstrate the use of poly(tartaric acid) in green synthesis of gold, silver, copper, and magnetite nanoparticles. Poly (tartaric acid) was used as both a reductant and coating agent. The obtained particles were analyzed by XRD, TEM/SEM, EDX, FTIR, and UV/VIS spectroscopy. Preliminary studies of the thermal behavior of mixtures of different types of particles with poly(tartaric acid) were also conducted. The topic is interesting and fits well with the scope of the journal. The results are significant and valuable. The authors made an effort to improve the manuscript, but there are still some issues that need to be addressed.

Question 1. While there are some additions, the introduction is still confusing and not focused enough. It requires substantial alterations. The novelty and motivation for the manuscript are also not apparent and should be more precise.

Answer 1. We are sorry that the introduction is still confusing. We thank the Reviewer for their comment. As a result of this observation, we have decided to rewrite the introduction of our work completely. We have tried to include more information about the novelty and motivation of our research in hopes that it will be clearer and ultimately improve its chances of the manuscript being accepted.

The results presentation and discussion are improved, but they are still confusing.

Question 2. Line 288: "as well as a certain amount of decarboxylation" – what does a certain amount of decarboxylation mean? It is not precise enough. Please rephrase.

Answer 2. The process of decarboxylation occurs minimally, and although the article published in 2019 (https://doi.org/10.1016/j.jclepro.2018.11.069) by our group explains it in detail, we have attempted to rephrase this paragraph in the revised version of the manuscript.

Question 3. Sections 3.1. to 3.4. are confused and should be carefully rewritten. The authors added some text, but these paragraphs should be rewritten entirely to be more straightforward. 

Answer 3. We made big efforts to rewrite carefully all sections mentioned by the referee, including the conclusion section.

Reviewer 2 Report (Previous Reviewer 2)

Comments and Suggestions for Authors

Thank you for the revised paper. Paper may be published in the present form. However, I recommend to clearly mention in abstract and keywords clearly types of copper nanoparticles for further better article search but it is for your consideration. Moreover, may be keywords gold, silver, copper should be mentioned better with word nanoparticle.

Comments on the Quality of English Language

Minor editing of English language required

Author Response

Dear Reviewer,

On behalf of the manuscript's co-authors with the code number ID: polymers-2727253, I would like to thank you for the detailed evaluation. We have carefully studied all the helpful remarks and made corrections, which we hope will meet your approval. We appreciate your efforts to provide valuable comments to improve the quality of our manuscript. Please find enclosed the revised manuscript wherein all comments are considered. As a result, of the reviewers’ suggestions, we have rewritten the introduction, large parts of the discussion section as well as the conclusion part. The changes are indicated using the Track Changes option in the revised manuscript. We believe that the paper has improved considerably and become more comprehensive and extended.

Thank you for the revised paper. Paper may be published in the present form.

Question 1. However, I recommend to clearly mention in abstract and keywords clearly types of copper nanoparticles for further better article search but it is for your consideration. Moreover, may be keywords gold, silver, copper should be mentioned better with word nanoparticle.

Answer 1. We appreciate the Reviewer's recommendations and have made the suggested changes to the abstract and keywords for the revised version of the manuscript.

Reviewer 3 Report (Previous Reviewer 3)

Comments and Suggestions for Authors

The authors have addressed the problem very well, and the manuscript can be accepted in the present form.

Comments on the Quality of English Language

Minor editing of English language required

Author Response

Dear Reviewer,

On behalf of the manuscript's co-authors with the code number ID: polymers-2727253, we would like to express our gratitude for your favourable response. Thank you for your time and effort in evaluating our work.  

Round 2

Reviewer 1 Report (Previous Reviewer 1)

Comments and Suggestions for Authors

The authors addressed all my comments. 

Comments on the Quality of English Language

Minor changes are required. 

This manuscript is a resubmission of an earlier submission. The following is a list of the peer review reports and author responses from that submission.

Round 1

Reviewer 1 Report

Comments and Suggestions for Authors

The manuscript entitled "Green synthesis of gold, silver, copper and magnetite particles using poly(tartaric acid) simultaneously as coating and reductant" aims to demonstrate the use of poly(tartaric acid) in green synthesis of gold, silver, copper, and magnetite nanoparticles. Poly (tartaric acid) was used as both a reductant and coating agent. The obtained particles were analyzed by XRD, TEM/SEM, EDX, FTIR, and UV/VIS spectroscopy. Preliminary studies of the thermal behavior of mixtures of different types of particles with poly(tartaric acid) were also conducted. The topic is interesting and fits well with the scope of the journal. The results are significant and valuable. Still, their presentation needs to be improved for the manuscript to be published. My comments are given below.

The abstract is suitable.

The introduction is confusing and not focused enough. It requires alterations. The novelty and motivation for the manuscript are also not clear and should be rewritten.

Materials and methods are well written and given in detail.

The results presentation and discussion are confusing.

Line 56: "All three types of nanoparticles are normally synthesized" – what does "normally" mean in this case? Please, rephrase.

Line 238: "as well as a small amount of decarboxylation" – what does a small amount of decarboxylation mean? Please,e rephrase.

Scheme 2 should be moved to the Supplementary material.

Sections 3.1. to 3.4. are confused and should be carefully rewritten.

The conclusions are in line with the results.

The literature is up to date.

Comments on the Quality of English Language

 Moderate editing of English language is required. 

Reviewer 2 Report

Comments and Suggestions for Authors

The authors presented the paper "Green synthesis of gold, silver, copper and magnetite particles using poly(tartaric acid) simultaneously as coating and reductant"

1) The reference list should be improved. More 2-3 years review papers should be cited in the Introduction section to show the progress in the area. I highly recommend not using references older than 10 years for all sections, except for historically important works. 

2) Introduction section should be improved. I highly recommend to insert 2022-2023 year references (review papers). The material about the nanoparticles' application is poor. Moreover, copper nanoparticles can be extremely toxic for the organism to the functions of copper in the body. Such nanoparticles usually don't stable and may influence pathologies. 

3) Why we should use poly(tartaric acid) is don't unclear. Has this acid unique property? If is not so important, why we need to synthesize various nanoparticles using it? I understand that improving stability of copper nanoparticles may be interesting. However, there are a number of works with gold, silver, and magnetite nanoparticles with various good coating. Why we need one more?

4) Have you proved poly(tartaric acid) structure in your work? I don't see any relevant data. If you have done it using literature method is not a reason not to do any characterization.

5) Gave you studied you nanoparticles by DLS? It is an important method of characterization in aqueous solution. By TEM and SEM I see high sized more than 100-150 nm which is not optimal for drug discovery. Moreover, nanoparticles form aggregates. Please, insert size distribution picture for TEM and SEM.

6) I am a bit confused about copper. Have you obtained copper or copper oxide (I) nanoparticles? It is an important difference. Pure stable copper nanoparticle synthesis is much harder work than copper oxide (I).

7) Have you studied the stability of your nanoparticles in water, salts, phosphate solution, various pH, oxygen presence for copper, cell media, etc.

8) FTIR spectra are not very. Why you have not good bands for organic compound?

9) The discussion is not clearly presented the data and the comparison between the results with the literature data. What is the new essential thing done in your work? 

10) For what application you assumed these nanoparticles? The properties of them must be adjusted to it. For example magnetite nanoparticles with the size of 100 nm may be useful for drug delivery. However, high-sized may be useful for magnetic separation.

11) Conclusion section is poor. The novelty and limitations of the work should be clearly mentioned in the Conclusion section and Abstract.

Comments on the Quality of English Language

Moderate editing of English language

Reviewer 3 Report

Comments and Suggestions for Authors

The authors report green synthesis of gold, silver, copper and magnetite particles using poly(tartaric acid) simultaneously as coating and reductant. The prepared nanoparticles were analyzed by XRD, TEM/SEM, EDX, FTIR and UV/VIS spectroscopy. The thermal behaviour of mixtures of different types of particles with poly(tartaric acid) were also conducted. The results are very interesting. However, some points of the manuscript should be improved. Specific comments are given below.

1.   The XPS is suggested to measure the samples to analyze the valence of chemical element. The reducing ability of poly(tartaric acid) is important.

2.   The coating behavior of poly(tartaric acid) is important, thus the stability of prepared nanoparticles is important. The hydrodynamic size of samples should be measured. The method is shown in the reference (International Journal of Biological Macromolecules. 2023;233:123513).

3.   The authors should summarize the static size of nanoparticles in a table.

4.   The molecular weight of poly(tartaric acid) should be measured by GPC. The NMR spectra of poly(tartaric acid) should be also measured.

5.   Please carefully check the manuscript for writing and grammar.

Comments on the Quality of English Language

Minor editing of English language required